# Compensation of Thermal Gradients Effects on a Quartz Crystal Microbalance

**DOI:** 10.3390/s23010024

**Published:** 2022-12-20

**Authors:** Marianna Magni, Diego Scaccabarozzi, Bortolino Saggin

**Affiliations:** 1Department of Mechanical Engineering Politecnico di Milano, Polo Territoriale di Lecco, Via G. Previati 1/c, 23900 Lecco, Italy; 2Rebel Dynamics, Via Carlo Porta 38, Cesana Brianza, 23861 Lecco, Italy

**Keywords:** QCM, TGA, CAM, calibration, uniform temperature, thermal gradient, frequency variation

## Abstract

Quartz Crystal Microbalances (QCM) are widely used instruments thanks to their stability, low mass, and low cost. Nevertheless, the sensitivity to temperature is their main drawback and is often a driver for their design. Though the crystal average temperature is mostly considered as the only disturbance, temperature affects the QCM measurements also through the in-plane temperature gradients, an effect identified in the past but mostly neglected. Recently, it has been shown that this effect can prevail over that of the average temperature in implementations where the heat for thermal control is released directly on the crystal through deposited film heaters. In this study, the effect of temperature gradients for this kind of crystal is analyzed, the sensitivity of frequency to the average temperature gradient on the electrode border is determined, and a correction is proposed and verified. A numerical thermal model of the QCM has been created to determine the temperature gradients on the electrode borders. The frequency versus temperature-gradient function has been experimentally determined in different thermal conditions. The correction function has been eventually applied to a QCM implementing a crystal of the same manufacturing lot as the one used for the characterization. The residual errors after the implementation of the correction of both average temperature and temperature gradients were always lower than 5% of the initial temperature disturbance. Moreover, using the correlation between the heater power dissipation and the generated temperature gradients, it has been shown that an effective correction strategy can be based on the measurement of the power delivered to the crystal without the determination of the temperature gradient.

## 1. Introduction

Quartz Crystal Microbalances are continuously expanding their application field because of the high and tunable sensitivity to mass deposition, stability in time, low cost, low power consumption, ease of reading, and capability of operating in very harsh environments, including high vacuum, cryogenic temperatures, and zero gravity [1,2]. Although QCMs have been used in many ways for a long time, the interest in this kind of sensor is still high because many different measurements can be done, either designing specific coatings selectively trapping specific chemical species [3,4,5,6] or using them to identify condensation or sublimation temperatures as in dew point sensors or for Thermo Gravimetric Analysis [7,8]. In most QCM applications, the crystal temperature is a relevant parameter, and, in many, it must be actively controlled to achieve the specific measurement goals. Temperature, nevertheless, is also the major disturbance in QCM measurements. Minimization of the temperature effects has been sought since the beginning of quartz crystals oscillators usage. The identification of the optimal crystal cut angle has been addressed [9,10], and configurations allowing for better temperature compensation are continuously investigated [11,12]. The average crystal temperature affects the crystal resonance; also, the temperature gradients within the crystal have similar effects [13,14,15]. Compensation of temperature effects based on the dual crystal configuration exploits the beating frequency of a reference crystal and the active one. The system is conceived to remove the effect of the average crystal temperature, assuming that both crystals have the same temperature [11,12], but the effect of temperature gradients is neglected. The uniform temperature condition is hardly achieved when the crystal is actively thermally controlled [16], and the gradient effect may become dominant when the heat is delivered directly on the crystal through film heaters deposited on it [17]. This work, starting from the findings reported in [17], derives the correlation between temperature gradient and frequency change, providing a relationship that can be used to compensate for the effect. An experimental campaign was performed on a QCM, measuring the resonance frequency in different thermal conditions. For each test condition, the temperature gradient was evaluated through a validated thermal model. The effect of the average crystal temperature was preliminarily removed, leaving the effect of the temperature gradient alone.

## 2. Temperature Gradient Determination from the Thermal Model

The thermal model of the QCM was developed to compute the temperature field on the crystal with the required accuracy and spatial resolution. The direct measurement of temperature through thermal mapping, in fact, was not accurate enough to derive the temperature gradient [17] because of the background disturbance, the low emissivity of the deposited electrodes, and the coarse spatial resolution of the infrared camera. As a consequence, the thermal model was implemented and preliminarily validated by correlation with the experimental tests described in [17]. The validated model has afterward been used to analyze crystals under different thermal conditions. The model geometry is shown in Figure 1.

A FE model based on the commercial software PTC CREO^®^ 9 was developed, representing the crystal and the mechanical mounting holding it. A relevant parameter for the modeling is the contact thermal resistance between the crystal and the supports. The contact was modeled as bonded surfaces for the contact area determined with the Hertzian contact model. The mounting thermal resistance, nevertheless, was one of the parameters for which experimental validation was mandatory. The materials used in the modeling were quartz for the crystal, AISI316 stainless steel for the supports, and a ceramic material, Macor^®^, for the support’s spacers. All materials were modeled as linear and isotropic. This is a simplification for the quartz crystal [18] that was adopted because, for the purposes of the analysis, the properties in the crystal plane are dominant. Table 1 reports the material properties used in the model; the AISI316 was in the CREO database, while the MACOR parameters were derived from the manufacturer datasheet and the quartz crystal ones from [14].

### 2.1. Analysis Conditions

To validate the model, the same conditions of the experimental tests (heating power and environmental temperature) were simulated. The reference analysis considered a power dissipation of the device of 0.445 W localized on the heater area at the crystal bottom surface. The total heat load was distributed considering the power density due to the film resistance shape. The power is non-uniformly distributed throughout the heater path, shaped with different film cross-sections to improve the temperature uniformity of the crystal. There are different power densities in three regions, as shown in Figure 2, according to the resistor shape.

### 2.2. Boundary Conditions

The thermal analysis is considered as boundary conditions for the convective and radiative exchange factors with the environment and the temperature of the spacers. The convection coefficient was computed considering the natural convection on a horizontal plate for the crystal [19]: (1)$$h_{conv}=C_{1}{\left(\frac{\Delta T}{M}\right)}^{1/4}$$
where Δ*T* is the difference between surface and environment temperatures, *M* is the area-to-perimeter ratio; *C*_1_ is a constant that for turbulent flow is 1.71 W m^−1.75^ K^−1.25^. The above equation was used for the crystal top surface, while the bottom corresponding value for downward-facing plates was exploited. The radiative exchanges make the model non-linear. However, both radiation and convection constraints were applied to the same surfaces with a linearization process. Starting from the equation:(2)$$W=\sigma \varepsilon A\left(T_{w}^{4}-T_{\infty }^{4}\right)$$
where *T_w_* is the temperature of the surface and *T*_∞_ the air one. After linearization it becomes:(3)$$W=h_{r}A\left(T_{w}-T_{\infty }\right)$$
where *h_r_* is the linearized radiation coefficient, given by:(4)$$h_{r}=\sigma \varepsilon \left(T_{w}^{2}+T_{\infty }^{2}\right)\left(T_{w}+T_{\infty }\right)$$

Eventually, the global heat exchange coefficient was obtained by adding the convection and the radiation coefficients:(5)$$h=h_{conv}+h_{r}$$

The last constraint imposed on the model was the temperatures of the supports’ spacer derived from the thermal images. In Figure 3, the values for the three supports are reported.

### 2.3. Results

Figure 4 shows both the temperature and the temperature gradient maps on the crystal determined with the thermal model. 

The plot in Figure 5 shows the temperature profile along the electrode edge. It can be noticed that the lower temperature value is reached on support 1. This is consistent with the observation that this area, the electrode pad, is the farthest from the heater. Considering the plot of Figure 6, the maximum gradient is between the supports 1–3 and 1–2, where there is the largest heat density. The minimum gradient on the electrode border is where there is no heat dissipation because the heater is interrupted because of the support 1 electrode pad. Furthermore, the thermal field is almost symmetric with respect to an axis passing through the center of the disk surface and contact 1, i.e., rotated 60° from the *x*-axis, as shown in Figure 7. It can be noticed that the hottest areas are on the heater but not in the regions at the highest power density, where the cooling of support 1 is prevailing. Concerning the crystal, the lowest temperature is about 68 °C, reached on the regions of contact with the supports, cooled by conduction. 

### 2.4. Model Validation

The thermal field predicted by the thermal analysis was compared with the one measured with a thermal mapper [17]. Five points of the electrode were identified on the model and compared with those corresponding to the thermal images. The same comparison was made for the heater region considering four points. Figure 8 and Figure 9 show the temperature maps, whereas Table 2 summarizes the prediction errors.

The temperatures measured during the test and those predicted by the thermal model are quite close: the maximum error on the electrode temperature is 1 °C, while the one on the heater is 0.34 °C. Considering that the uncertainty in the electrode temperature measurement was 0.85 °C and on the heater 0.42 °C [17], the contribution of the model uncertainty is of the order of 0.5 °C, a value considered acceptable for this study. The above comparison, therefore, validated the numerical model.

## 3. Temperature Gradient Sensitivity

To correlate the temperature gradient with the frequency variation, different thermal loads and boundary conditions used during the experimental activity were simulated, assessing the temperature gradients as a result. The different gradients were obtained by feeding the heater with different powers and keeping constant the other conditions. Generating different temperature gradients while maintaining the same average temperature would have been quite complex, requiring the contemporary application of heating and cooling on different crystal regions. Thus, the resulting set of conditions was characterized by different temperature gradients and different average temperatures. Assuming that the effect of the temperature gradients and that of the average temperature can be superimposed, the dependency of the frequency on the temperature gradient was determined by preliminarily removing the effect of the average temperature. The correction for the average temperature was based on the crystal characterization reported in [17], where the behavior of the crystal in a uniform temperature environment was assessed and discussed.

Low-temperature tests, below 50 °C, were eventually discarded because of the interfering effect of contaminants condensation on the crystal.

The thermal gradients were evaluated for different power dissipations of the heaters and used to assess the effects on the crystal frequency are shown in Figure 10.

The average thermal gradient was correlated with the frequency change after the application of the correction for the average temperature. The results are reported in Figure 11. 

The correlation coefficient for the thermal gradient versus frequency fitting with a 3rd-order polynomial was 0.964 with a root mean square error of 55 Hz. Eventually, the effectiveness of the correction procedure was verified with a new experimental campaign on a standard QCM, including a crystal of the same manufacturing lot as the one used for characterization. By using the model of the thermal gradient effects, i.e., the polynomial regression of the gradient values of Figure 11, along with that of the uniform temperature, the frequency variation of the crystal with more general temperature fields can be predicted. To validate the model, the resonance frequency was measured while supplying different powers to the crystal resistor. Thermal analyses were performed to compute the corresponding gradient, and the corresponding corrective factor was evaluated. The average temperature on the crystal border was measured with the RTD deposited on the surface of the crystal and compared with the thermal model predictions, verifying that the differences were within the already identified range, i.e., within ±0.5 °C.

The average temperature was then used to compute the corresponding frequency change under uniform temperature. Eventually, by adding the frequency variation due to the thermal gradient with the previous one, the predicted frequency variation was obtained. The predicted frequency, along with the measured one, and the effect of the uniform temperature are reported in Figure 12.

It can be noticed that at low temperatures (between 40 °C and 50 °C), the correction significantly overestimates the measured frequency change. This has been attributed to the condensation of contaminants that was also observed in other low-temperature tests [17], so these points were not considered in the assessment of the model error. With the crystal temperature increasing, the frequency change becomes more relevant, but it is still quite accurately predicted by the model, allowing it to correct for the effect of the gradient with residual errors smaller than 5%.

The above procedure relies on the thermal model to determine the temperature gradient on the crystal. Nevertheless, considering that the heat flux is proportional to the temperature gradient, a correlation between the average gradient on the electrode border and the power dissipated by the heater deposited right there is expected. The mean thermal gradients, computed while performing five heating cycles, were plotted against the heating power in Figure 13. The correlation is evident from the linear plot. Moreover, the least square fitting leads to a linearity standard deviation of 0.027 [°C/mm]. The latter figure would be the standard uncertainty of the temperature gradient when it was determined from the power dissipation. The propagation of this contribution on the correction function leads to a negligible contribution in comparison with the observed 5% error. Therefore, the correction procedure based directly on the measurement of the heating power performs with the same accuracy as the one shown above, based on the evaluation of the temperature gradient but without requiring the implementation of the crystal thermal analysis.

## 4. Conclusions

The full characterization of the temperature effects on QCMs must investigate along with the effect of the average temperature, the one related to the temperature gradients. In some crystal configurations, like those recently proposed with built-in film heaters and temperature sensors, the temperature gradients may have a dominant effect. In this study, the relationship between the average temperature gradient on the electrode border and the frequency change was experimentally determined for QCM crystals of this type. The temperature gradient was determined for different test conditions through a specifically developed and validated thermal model. The modeling of both effects of temperature, i.e., the average and the spatial gradients, was performed for temperatures of the crystal between the ambient one and 100 °C and with average gradient amplitudes from 2 to 8 °C/mm.

The correction procedure of the temperature effects based on this model allowed for achieving residual errors below 5% for crystals temperatures above 50 °C. More significant discrepancies were observed at lower temperatures of the crystal. Nevertheless, in these conditions, the condensation of contaminants on the QCM jeopardizes the testing repeatability. Testing under a high vacuum is expected to improve the repeatability also in this condition; nevertheless, it is less interesting because the effect is not very large. Although the procedure has been developed using the temperature gradient determined from the thermal model of the crystal, it was additionally shown that evaluating the temperature gradient through the correlation with the power dissipation of the heaters allows for performing the correction without significant accuracy loss. This implementation strongly reduces the complexity and the effort required for the correction because all the needed information is directly available from the standard QCM monitoring parameters. This study allows for an easy correction of the temperature gradient effect when the environmental temperature is close to 20 °C; QCM is often operating at cryogenic temperatures, and extending the characterization in these conditions is a natural extension. Considering that reducing the disturbance source would ease any correction procedure, the other evolution of this research will be in the direction of changing the crystal design to reduce the temperature disturbance. This can be achieved by optimizing the power distribution on the crystal, with a proper shaping of the deposited film heater, for the purpose of minimizing the temperature gradients.

## Figures and Tables

**Figure 1 sensors-23-00024-f001:**
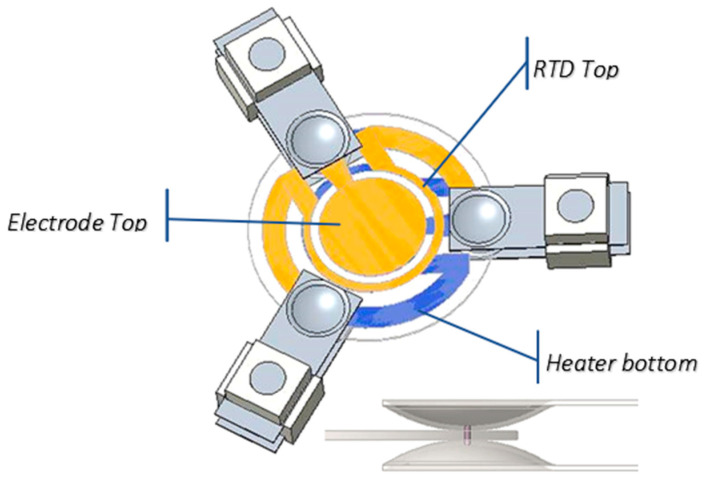
Model of crystal and mounting system with detail of the mounting contacts.

**Figure 2 sensors-23-00024-f002:**
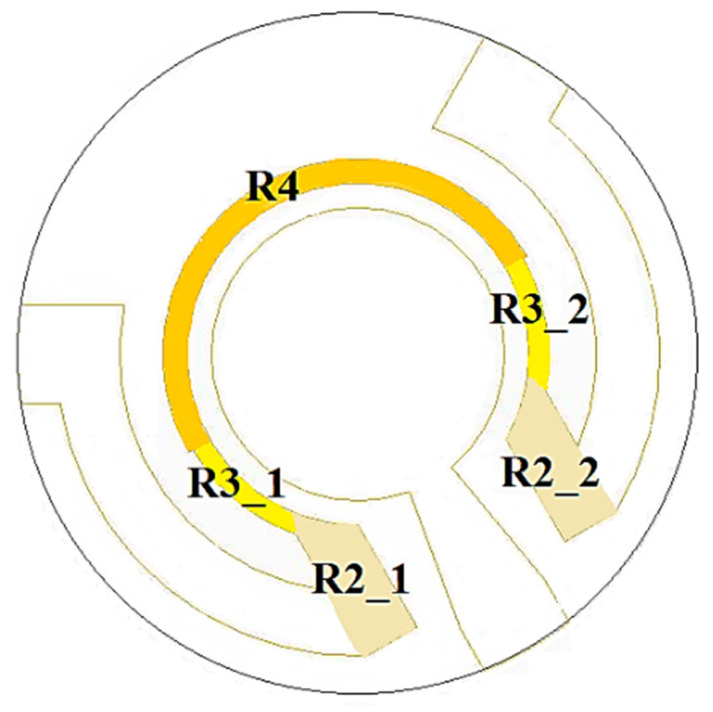
Different heater areas for power distribution.

**Figure 3 sensors-23-00024-f003:**
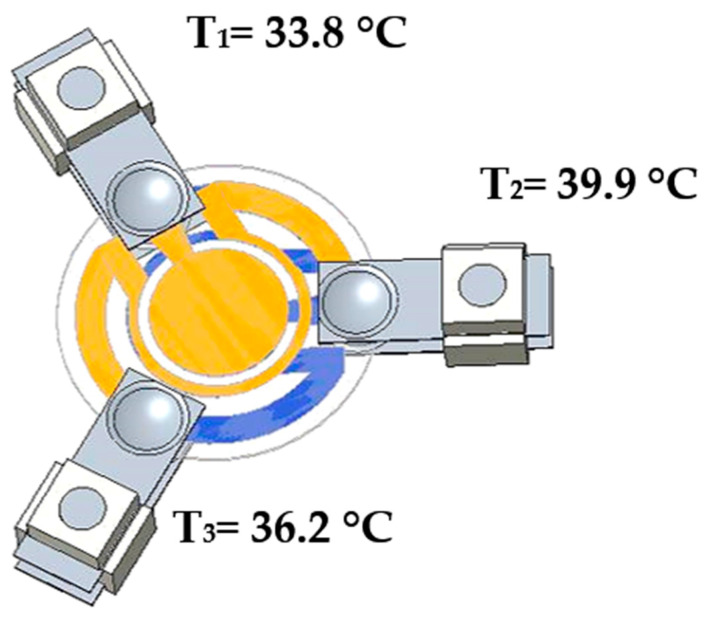
Temperatures of the supports’ spacers.

**Figure 4 sensors-23-00024-f004:**
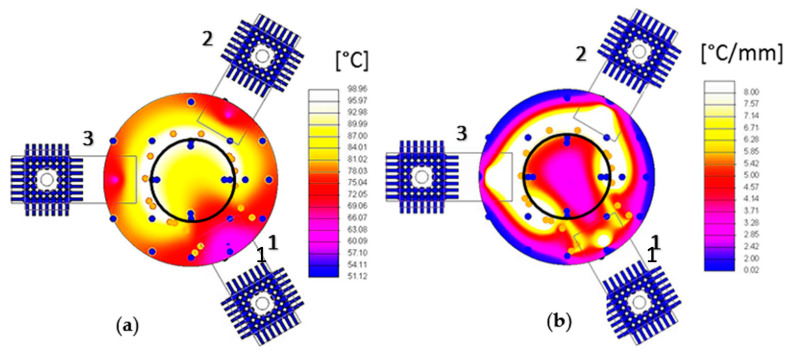
(**a**) Electrode temperature mapping; (**b**) Temperature gradient.

**Figure 5 sensors-23-00024-f005:**
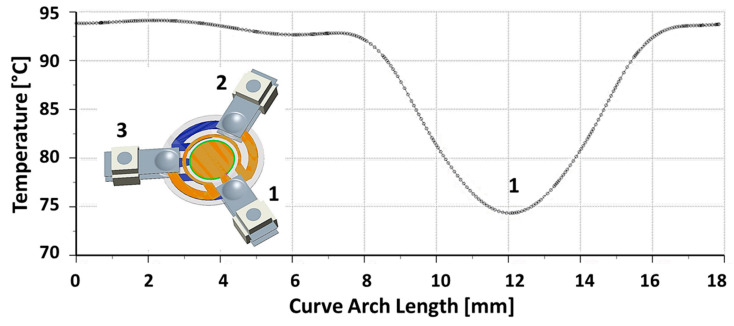
Temperature profile on the electrode edge.

**Figure 6 sensors-23-00024-f006:**
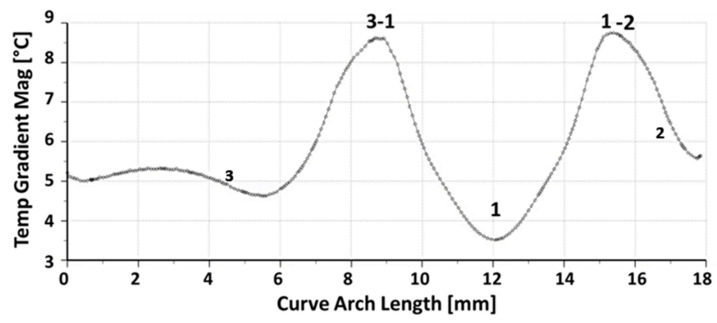
Temperature gradient magnitude on the electrode edge.

**Figure 7 sensors-23-00024-f007:**
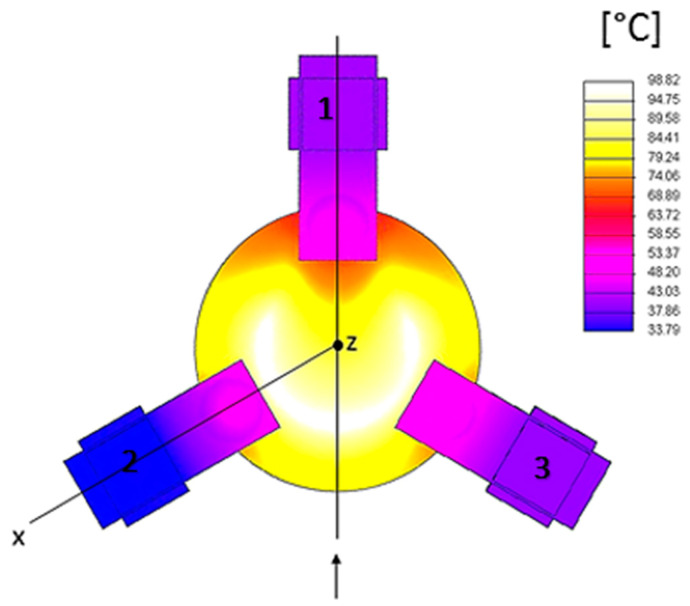
Thermal field of the complete model.

**Figure 8 sensors-23-00024-f008:**
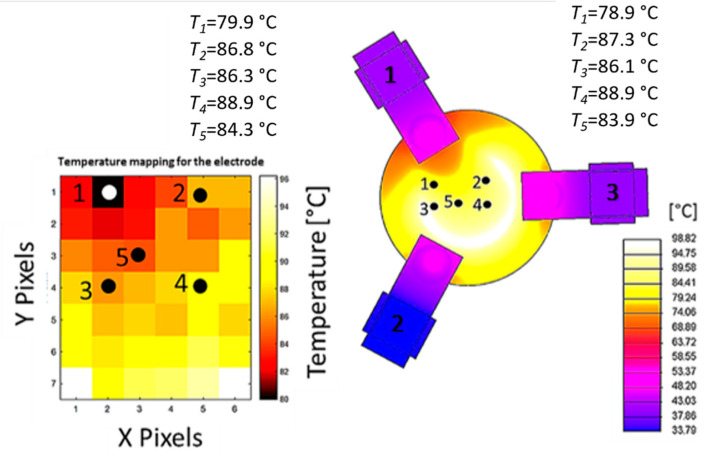
Temperature map on the electrode: comparison between the measured thermal image (**left**) and the model predictions (**right**).

**Figure 9 sensors-23-00024-f009:**
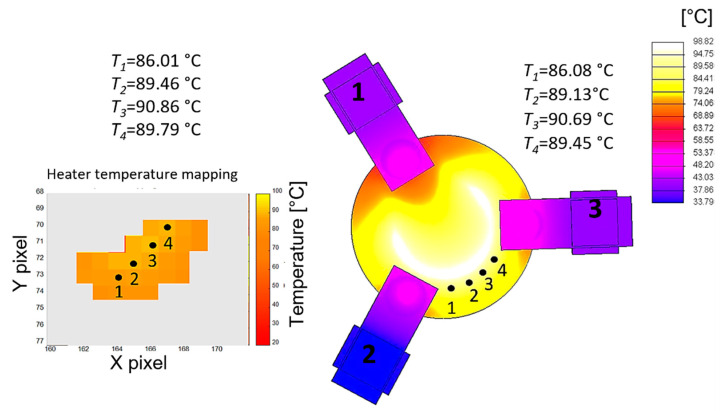
Temperature map on the heater: comparison between the measured thermal image (**left**) and the model predictions (**right**).

**Figure 10 sensors-23-00024-f010:**
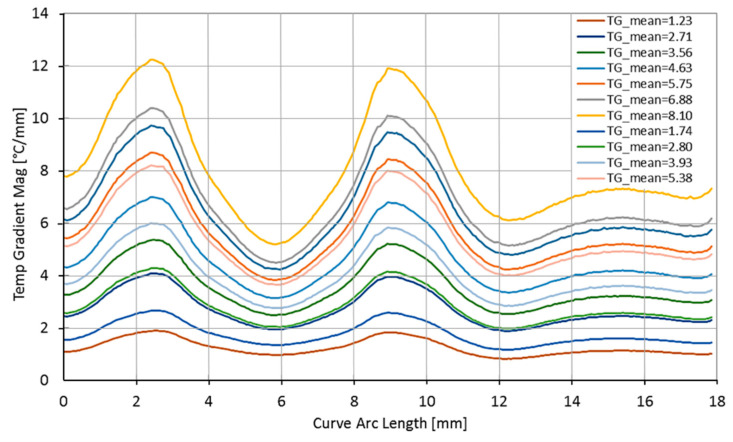
The average thermal gradient (TG_ mean) on the electrode edge.

**Figure 11 sensors-23-00024-f011:**
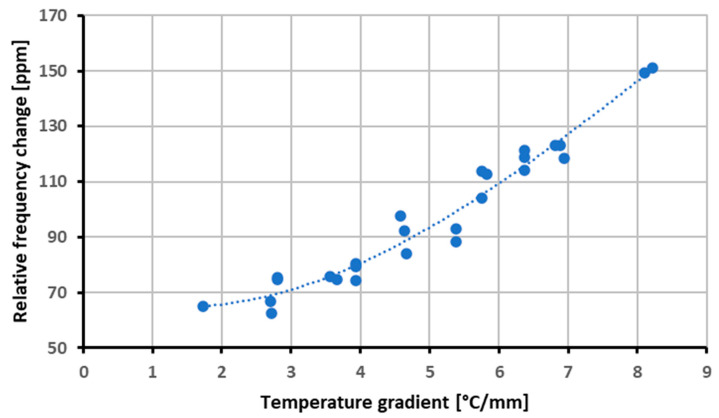
Frequency variation vs. thermal gradient.

**Figure 12 sensors-23-00024-f012:**
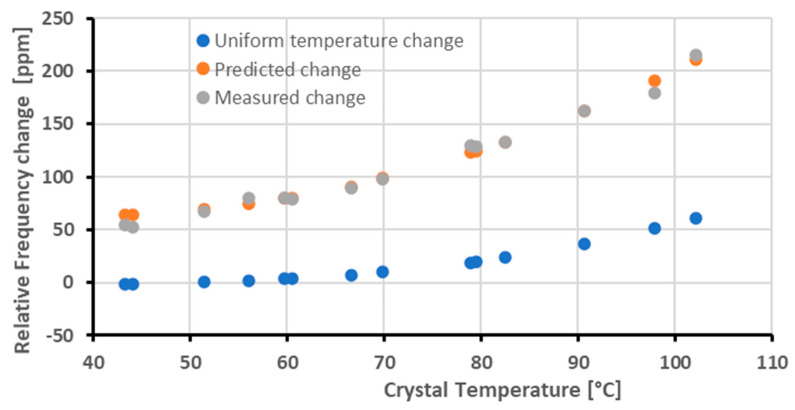
Comparison between the relative frequency changes predicted by the model and the measured ones.

**Figure 13 sensors-23-00024-f013:**
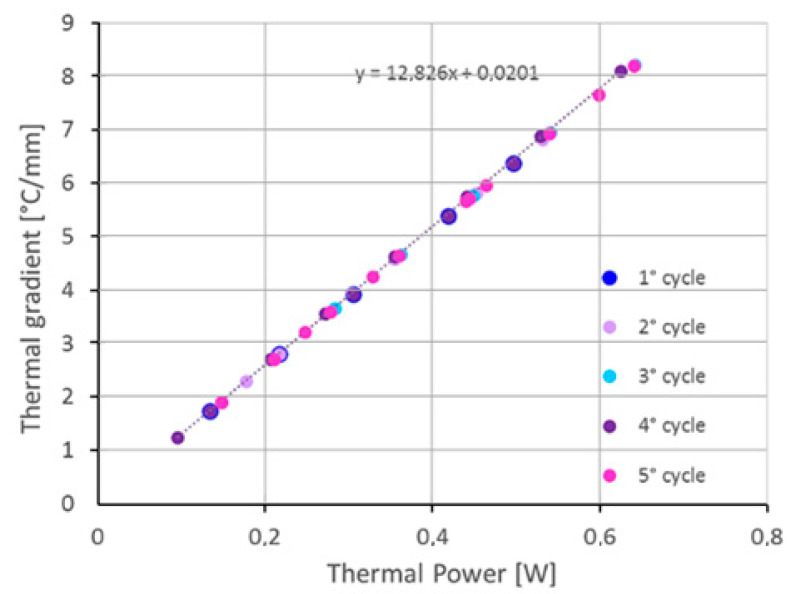
Temperature gradient vs. power dissipation.

**Table 1 sensors-23-00024-t001:** Material properties used in the model.

Components	Density	Young Modulus	Poisson Coefficient	CTE	Thermal Conductivity
	[kg/m^3^]	[MPa]		[1/°C]	[W/(m °C)]
QUARTZ	2649	76,500	0.17	7.10 × 10^−6^	6.2
AISI316	8000	193,000	0.29	1.60 × 10^−5^	14
MACOR	2520	66,900	0.29	9.00 × 10^−6^	1.5

**Table 2 sensors-23-00024-t002:** Model errors: the difference between model-predicted temperatures and measured ones on the selected points on the electrode and heater.

Electrode Points	Error [°C]	Heater Points	Error [°C]
**1**	1	**1**	−0.07
**2**	−0.5	**2**	0.33
**3**	0.2	**3**	0.17
**4**	0	**4**	0.34
**5**	0.4

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
