# Peer review of "Compensation of Thermal Gradients Effects on a Quartz Crystal Microbalance"

_sensors, 2022, doi:10.3390/s23010024_

Round 1

Reviewer 1 Report

In this paper, the authors systematically investigated the effect of temperature gradients of QCM measurements. The authors studied the sensitivity of frequency to the average temperature gradient on the electrode border. In general, this is an interesting study. Below are some question and comments that need to be addressed.

1. The model is based on three mounting systems. How could the model be generalized with more than three or less than three mounting systems?

2. How would the model behave with smaller temperature gradient and lower initial temperature?

3. What is the maximum thermal power, temperature, and thermal gradient such that the model still holds?

4. The resolution of all figures could be improved. Figure 12-15 looks like directly come from excel. It need to be made into more professional format for publishing purpose.

Author Response

Authors thank the reviewer for the helpful suggestions, in the following the reply to the specific comments.

In this paper, the authors systematically investigated the effect of temperature gradients of QCM measurements. The authors studied the sensitivity of frequency to the average temperature gradient on the electrode border. In general, this is an interesting study. Below are some question and comments that need to be addressed.

  1. The model is based on three mounting systems. How could the model be generalized with more than three or less than three mounting systems?

The gradient issue is a general problem that is strictly depending on the crystal configuration, our study has a general validity but the correction functions must be derived for the specific implementation and for each crystal manufacturing batch

  1. How would the model behave with smaller temperature gradient and lower initial temperature?

The sensitivity to gradients increases with the gradient magnitude so, small gradients seems not to influence significantly the QCM, concerning a possible combined effect of temperature and gradient this has still to be investigated. We assumed the two effects as independent because with the temperature changes due to the ambient this seems the case nevertheless, an investigation under cryogenic conditions, that is a common operating mode of QCMs, is a foreseen future development.

  1. What is the maximum thermal power, temperature, and thermal gradient such that the model still holds?

In our case the model has been derived for power up to 0.7 W and temperature gradients up to 12 °C/mm

  1. The resolution of all figures could be improved. Figure 12-15 looks like directly come from excel. It need to be made into more professional format for publishing purpose.

Pictures have been improved

Reviewer 2 Report

The authors et al. report the effect of compensating for thermal gradients in quartz crystal microbalance and determine the sensitivity of frequency to the average temperature gradient at the electrode boundaries. This work will be helpful to researchers in related fields. Therefore, after addressing the following issues, this manuscript seems worthy of publication in the journal Sensors.

1.  Page 1, line 8. Add the abbreviation QCM after the Quartz Crystal Microbalance. Abbreviations must be entered on the page where they are first mentioned, included in the abstract.

2.  This paper and reference 17 have confirmed the existence of temperature gradients experimentally, but (Fig. 1) the D-model of the crystal and mounting system does not label every component, please label it separately. and add the actual plot (as in ref. 17 (Fig. 7) for the measurement setup of the microbalance calibration).

3.  It is suggested a more description on the QCM experiments.

4.  In addition, the authors are advised to utilize electron microscopy (SEM, AFM, et.) for morphological and structural analysis, which will contribute to a comprehensive understanding of the fundamentals of sensor behavior and should be the continuation topic of this research.

5.  Figures 4, 7, 8 and 9 do not mark components. Please label components 1, 2 and 3 according to Figure 5 for the convenience of readers.

6.  Please label the x and y axis major tick marks uniformly (style out or in). In addition, Figures 13 and 15 have no major tick marks. Please reshape it.

7.  Is there a qualitative or quantitative relationship between frequency changes and thermal gradients? How many experiments for each point and what is the Y error? There is no statement or data in the text, please reshape it.

8.  What is the repeatability of the measurements? (Experiments with gold-coated crystals made from the same batch.)

9.  The author successfully publishes in the paper the effect of compensating thermal gradients on quartz crystal microbalance. Therefore, the authors should emphasize in the conclusion what are the future application directions of this study?

Author Response

Authors thank the reviewer for the helpful suggestions. In the following the reply to the specific issues.

The authors et al. report the effect of compensating for thermal gradients in quartz crystal microbalance and determine the sensitivity of frequency to the average temperature gradient at the electrode boundaries. This work will be helpful to researchers in related fields. Therefore, after addressing the following issues, this manuscript seems worthy of publication in the journal Sensors.

  1. Page 1, line 8. Add the abbreviation QCM after the Quartz Crystal Microbalance. Abbreviations must be entered on the page where they are first mentioned, included in the abstract.

QCM abbreviation has been introduced from the abstract

  1. This paper and reference 17 have confirmed the existence of temperature gradients experimentally, but (Fig. 1) the D-model of the crystal and mounting system does not label every component, please label it separately. and add the actual plot (as in ref. 17 (Fig. 7) for the measurement setup of the microbalance calibration).

Labels have been added

  1. It is suggested a more description on the QCM experiments.

The description has been reformulated hopefully it is now clearer

  1. In addition, the authors are advised to utilize electron microscopy (SEM, AFM, et.) for morphological and structural analysis, which will contribute to a comprehensive understanding of the fundamentals of sensor behavior and should be the continuation topic of this research.

A study on the deposited film is ongoing SEM analysis is foreseen in this context

  1. Figures 4, 7, 8 and 9 do not mark components. Please label components 1, 2 and 3 according to Figure 5 for the convenience of readers.

Labels have been added

  1. Please label the x and y axis major tick marks uniformly (style out or in). In addition, Figures 13 and 15 have no major tick marks. Please reshape it.

Plots styles have been changed

  1. Is there a qualitative or quantitative relationship between frequency changes and thermal gradients? How many experiments for each point and what is the Y error? There is no statement or data in the text, please reshape it.

Maybe we do not understand the remark, Figure 13 shows the frequency-temp.gradient relationship, each point is a test, the error RMSE is reported in the test though not in the plot.

  1. What is the repeatability of the measurements? (Experiments with gold-coated crystals made from the same batch.)

The repeatability for each test condition i.e. a given power of the heater, is similar to the model error i.e. about 50 Hz but, it includes the effect of different electrode average temperatures as results of the environmental temperature changes and the model accounts for that.

  1. The author successfully publishes in the paper the effect of compensating thermal gradients on quartz crystal microbalance. Therefore, the authors should emphasize in the conclusion what are the future application directions of this study?

The future developments of the study have been added to the conclusions.

Reviewer 3 Report

Dear Authors,

The first impression is the prepared manuscript should follow the template of the Journal of SENSORS. The submitted version is typewritten by Microsoft Word, doing revisions!

Kindly consider the following:

1-Abstract: Line 9: low cost. Nevertheless, the sensitivity to temperature... should be rewritten.

2- Line 15: in this paper .... the more academic sound should be present for the manuscripts submitted to the journals of MDPI publisher. It can rewritten: in the current study, in the current research, ....

3-The Abstract can be conciser and rewritten with easier follow.

4-Introduction Section: Quartz Crystal Microbalances are continuously ...can be rewritten: Quartz Crystal Microbalances (QCM) are continuously...or the term QCM shall be abbreviated in the Abstract!

5-Line 33: of the high sensitivity and tunable sensitivity to mass.... because of the high and tunable sensitivity....

6-Line 36: been used in many ways for a long time, the interest in this kind....

7-from line 35 to line 40, the purpose is not clear enough.

8-Line 48: dual crystal as in [11, 12] is conceived to remove the effect ...the expression is not clear as it should be clarified by authors then cite these two references.

9-Line 50 is not clear. thermally controlled...

10:Line 55: To do that an experimental.... can be rewritten as: To achieve this objective, an experimental...

II Temperature Gradient Determination from the Thermal Model

11-Line 63: enough to derive reliable the temperature gradient,, can be rewritten: enough to derive a reliable temperature gradient...As a consequence....

12-Which kind of software is used?

13- what is D model in Figure 1?

14-Table 3, reference?

15-Line 83 & 84: (heating power and environmental temperature)

16-in  the Analysis conditions subsection, it is not clear how the authors made this assumption?

17-Figure 4, it should be specified Figure 4 a for temperature distribution and Figure 4 b for gradient temperature distribution.

General Remarks before continuing the reviewing process:

Overall, my suggestion for the authors is to rewrite the manuscript carefully to be submitted again for a helpful explanation and to be easier to follow up. English editing for this manuscript is essential. Paying attention to punctuation and sentence arrangement because sometimes the way of representation is several parts throughout this manuscript leads to misunderstanding!

Author Response

Authors thank the reviewer for the accurate revision and the helpful suggestions, in the following the reply to the specific remarks.

Dear Authors,

The first impression is the prepared manuscript should follow the template of the Journal of SENSORS. The submitted version is typewritten by Microsoft Word, doing revisions!

Kindly consider the following:

1-Abstract: Line 9: low cost. Nevertheless, the sensitivity to temperature... should be rewritten.

Corrected.

2- Line 15: in this paper .... the more academic sound should be present for the manuscripts submitted to the journals of MDPI publisher. It can rewritten: in the current study, in the current research, ....

Modified as suggested

3-The Abstract can be conciser and rewritten with easier follow.

The abstract (and the paper on the whole) has been more thoroughly reviewed and partially rewritten.

4-Introduction Section: Quartz Crystal Microbalances are continuously ...can be rewritten: Quartz Crystal Microbalances (QCM) are continuously...or the term QCM shall be abbreviated in the Abstract!

The acronym has been introduced in the abstract

5-Line 33: of the high sensitivity and tunable sensitivity to mass.... because of the high and tunable sensitivity....

Corrected

6-Line 36: been used in many ways for a long time, the interest in this kind....

Corrected

7-from line 35 to line 40, the purpose is not clear enough.

The sentence has been rephased hopefully making it clear

8-Line 48: dual crystal as in [11, 12] is conceived to remove the effect ...the expression is not clear as it should be clarified by authors then cite these two references.

The sentence has been re-phrase

9-Line 50 is not clear. thermally controlled...

The sentence has been re-phrased

10:Line 55: To do that an experimental.... can be rewritten as: To achieve this objective, an experimental...

II Temperature Gradient Determination from the Thermal Model

11-Line 63: enough to derive reliable the temperature gradient, can be rewritten: enough to derive a reliable temperature gradient...As a consequence....

12-Which kind of software is used?

Reference has been included

13- what is D model in Figure 1?

Typo, removed

14-Table 3, reference?

References have been included

15-Line 83 & 84: (heating power and environmental temperature)

Corrected

16-in  the Analysis conditions subsection, it is not clear how the authors made this assumption?

For model validation the selected power dissipation was an intermediate value and not the maximum one to avoid the condition of high temperature where the (more predictable) radiative exchanges are dominant with the risk of masking the modeling errors in the conductive paths.

17-Figure 4, it should be specified Figure 4 a for temperature distribution and Figure 4 b for gradient temperature distribution.

Labels a) and  b) have been added.

General Remarks before continuing the reviewing process:

Overall, my suggestion for the authors is to rewrite the manuscript carefully to be submitted again for a helpful explanation and to be easier to follow up. English editing for this manuscript is essential. Paying attention to punctuation and sentence arrangement because sometimes the way of representation is several parts throughout this manuscript leads to misunderstanding!

Authors apologize for the many language errors, the paper has been more thoroughly checked and partially rewritten.

Round 2

Reviewer 3 Report

these modifications are accepted

Author Response

Authors thank the reviewer.